# Sparse and Low-Rank Tensor Decomposition

**Parikshit Shah**
parikshit@yahoo-inc.com

**Nikhil Rao**
nikhilr@cs.utexas.edu

**Gongguo Tang**
gtang@mines.edu

## Abstract

Motivated by the problem of robust factorization of a low-rank tensor, we study the question of sparse and low-rank tensor decomposition. We present an efficient computational algorithm that modifies Leurgans' algoirthm for tensor factorization. Our method relies on a reduction of the problem to sparse and low-rank matrix decomposition via the notion of tensor contraction. We use well-understood convex techniques for solving the reduced matrix sub-problem which then allows us to perform the full decomposition of the tensor. We delineate situations where the problem is recoverable and provide theoretical guarantees for our algorithm. We validate our algorithm with numerical experiments.

## 1 Introduction

Tensors are useful representational objects to model a variety of problems such as graphical models with latent variables [1], audio classification [20], psychometrics [8], and neuroscience [3]. One concrete example proposed in [1] involves topic modeling in an exchangeable bag-of-words model wherein given a corpus of documents one wishes to estimate parameters related to the different topics of the different documents (each document has a unique topic associated to it). By computing the empirical moments associated to (exchangeable) bi-grams and tri-grams of words in the documents, [1] shows that this problem reduces to that of a (low rank) tensor decomposition. A number of other machine learning tasks, such as Independent Component Analysis [11], and learning Gaussian mixtures [2] are reducible to that of tensor decomposition. While most tensor problems are computationally intractable [12] there has been renewed interest in developing tractable and principled approaches for the same [4, 5, 12, 15, 19, 21, 24–27].

In this paper we consider the problem of performing tensor decompositions when a subset of the entries of a low-rank tensor $X$ are corrupted adversarially, so that the tensor observed is $Z = X + Y$ where $Y$ is the corruption. One may view this problem as the tensor version of a sparse and low-rank matrix decomposition problem as studied in [6, 9, 10, 13]. We develop an algorithm for performing such a decomposition and provide theoretical guarantees as to when such decomposition is possible. Our work draws on two sets of tools: (a) The line of work addressing the Robust PCA problem in the matrix case [6, 9], and (b) Application of Leaurgans' algorithm for tensor decomposition and tensor inverse problems [4, 17, 24].

Our algorithm is computationally efficient and scalable, it relies on the key notion of tensor contraction which effectively reduces a tensor problem of dimension $n \times n \times n$ to four decompostion problems for matrices of size $n \times n$. One can then apply convex methods for sparse and low-rank matrix decomposition followed by certain linear algebraic operations to recover the constituent tensors. Our algorithm not only produces the correct decomposition of $Z$ into $X$ and $Y$, but also produces the low rank factorization of $X$. We are able to avoid tensor unfolding based approaches [14, 21, 26] which are expensive and would lead to solving convex problems that are larger by orders of magnitude; in the $3^{rd}$ order case the unfolded matrix would be $n^2 \times n$. Furthermore, our method is

conceptually simple, to impelement as well as to analyze theoretically. Finally our method is also modular – it can be extended to the higher order case as well as to settings where the corrupted tensor $\boldsymbol{Z}$ has missing entries, as described in Section 5.

## 1.1 Problem Setup

In this paper, vectors are denoted using lower case characters (e.g. $x, y, a, b$, etc.), matrices by upper-case characters (e.g. $X, Y$, etc.) and tensors by upper-case bold characters (e.g. $\boldsymbol{X}, \boldsymbol{T}, \boldsymbol{A}$ etc.). We will work with tensors of third order (representationally to be thought of as three-way arrays), and the term mode refers to one of the axes of the tensor. A slice of a tensor refers to a two dimensional matrix generated from the tensor by varying indices along two modes while keeping the third mode fixed. For a tensor $\boldsymbol{X}$ we will refer to the indices of the $i^{th}$ mode-1 slice (i.e., the slice corresponding to the indices $\{i\} \times [n_2] \times [n_3]$) by $S_i^{(1)}$, where $[n_2] = \{1, 2, \ldots, n_2\}$ and $[n_3]$ is defined similarly. We denote the matrix corresponding to $S_i^{(1)}$ by $X_i^1$. Similarly the indices of the $k^{th}$ mode-3 slice will be denoted by $S_k^{(3)}$ and the matrix by $X_k^3$.

Given a tensor of interest $\boldsymbol{X}$, consider its decomposition into rank one tensors

$$\boldsymbol{X} = \sum_{i=1}^{r} \lambda_i u_i \otimes v_i \otimes w_i, \tag{1}$$

where $\{u_i\}_{i=1,\ldots,r} \subseteq \mathbb{R}^{n_1}$, $\{v_i\}_{i=1,\ldots,r} \subseteq \mathbb{R}^{n_2}$, and $\{w_i\}_{i=1,\ldots,r} \subseteq \mathbb{R}^{n_3}$ are unit vectors. Here $\otimes$ denotes the tensor product, so that $\boldsymbol{X} \in \mathbb{R}^{n_1 \times n_2 \times n_3}$ is a tensor of order 3 and dimension $n_1 \times n_2 \times n_3$. Without loss of generality, throughout this paper we assume that $n_1 \leq n_2 \leq n_3$. We will present our results for third order tensors, and analogous results for higher orders follow in a transparent manner. We will be dealing with *low-rank* tensors, i.e. those tensors with $r \leq n_1$. Tensors can have rank larger than the dimension, indeed $r \geq n_3$ is an interesting regime, but far more challenging and is a topic left for future work.

Kruskal's Theorem [16] guarantees that tensors satisfying Assumption 1.1 below have a unique minimal decomposition into rank one terms of the form (1). The number of terms is called the (Kruskal) rank.

**Assumption 1.1.** $\{u_i\}_{i=1,\ldots,r} \subseteq \mathbb{R}^{n_1}$, $\{v_i\}_{i=1,\ldots,r} \subseteq \mathbb{R}^{n_2}$, and $\{w_i\}_{i=1,\ldots,r} \subseteq \mathbb{R}^{n_3}$ are sets of linearly independent vectors.

While rank decomposition of tensors in the worst case is known to be computationally intractable [12], it is known that the (mild) assumption stated in Assumption 1.1 above suffices for an algorithm known as Leurgans' algorithm [4, 18] to correctly identify the factors in this unique decomposition. In this paper, we will make this assumption about our tensor $\boldsymbol{X}$ throughout. This assumption may be viewed as a "genericity" or "smoothness" assumption [4].

In (1), $r$ is the rank, $\lambda_i \in \mathbb{R}$ are scalars, and $u_i \in \mathbb{R}^{n_1}, v_i \in \mathbb{R}^{n_2}, w_i \in \mathbb{R}^{n_3}$ are the tensor factors. Let $U \in \mathbb{R}^{n_1 \times r}$ denote the matrix whose columns are $u_i$, and correspondingly define $V \in \mathbb{R}^{n_2 \times r}$ and $W \in \mathbb{R}^{n_3 \times r}$. Let $\boldsymbol{Y} \in \mathbb{R}^{n_1 \times n_2 \times n_3}$ be a sparse tensor to be viewed as a "corruption" or adversarial noise added to $\boldsymbol{X}$, so that one observes:

$$\boldsymbol{Z} = \boldsymbol{X} + \boldsymbol{Y}.$$

The problem of interest is that of decomposition, i.e. recovering $\boldsymbol{X}$ and $\boldsymbol{Y}$ from $\boldsymbol{Z}$.

For a tensor $\boldsymbol{X}$, we define its mode-3 *contraction* with respect to a contraction vector $a \in \mathbb{R}^{n_3}$, denoted by $X_a^3 \in \mathbb{R}^{n_1 \times n_2}$, as the following matrix:

$$\left[X_a^3\right]_{ij} = \sum_{k=1}^{n_3} \boldsymbol{X}_{ijk} a_k, \tag{2}$$

so that the resulting $n_1 \times n_2$ matrix is a weighted sum of the mode-3 slices of the tensor $\boldsymbol{X}$. Under this notation, the $k^{th}$ mode-3 slice matrix $X_k^3$ is a mode-3 contraction with respect to the $k^{th}$ canonical basis vector. We similarly define the mode-1 contraction with respect to a vector $c \in \mathbb{R}^{n_1}$ as

$$\left[X_c^1\right]_{jk} = \sum_{i=1}^{n_1} \boldsymbol{X}_{ijk} c_i. \tag{3}$$

In the subsequent discussion we will also use the following notation. For a matrix $M$, $\|M\|$ refers to the spectral norm, $\|M\|_*$ the nuclear norm, $\|M\|_1 := \sum_{i,j}|M_{ij}|$ the elementwise $\ell_1$ norm, and $\|M\|_\infty := \max_{i,j}|M_{i,j}|$ the elementwise $\ell_\infty$ norm.

## 1.2 Incoherence

The problem of sparse and low-rank decomposition for matrices has been studied in [6, 9, 13, 22], and it is well understood that exact decomposition is not always possible. In order for the problem to be identifiable, two situations must be avoided: (a) the low-rank component $X$ must not be sparse, and (b) the sparse component $Y$ must not be low-rank. In fact, something stronger is both necessary and sufficient: the tangent spaces of the low-rank matrix (with respect to the rank variety) and the sparse matrix (with respect to the variety of sparse matrices) must have a *transverse* intersection [9].

For the problem to be amenable to recovery using comptationally tractable (convex) methods, somewhat stronger, *incoherence* assumptions are standard in the matrix case [6,7,9]. We will make similar assumptions for the tensor case, which we now describe.

Given the decomposition (1) of $X$ we define the following subspaces of matrices:

$$
\begin{aligned}
T_{U,V} &= \left\{ UA^T + BV^T \ : \ A \in \mathbb{R}^{n_2 \times r}, B \in \mathbb{R}^{n_1 \times r} \right\} \\
T_{V,W} &= \left\{ VC^T + DW^T \ : \ C \in \mathbb{R}^{n_3 \times r}, D \in \mathbb{R}^{n_2 \times r} \right\}.
\end{aligned}
\tag{4}
$$

Thus $T_{U,V}$ is the set of rank $r$ matrices whose column spaces are contained in $\mathrm{span}(U)$ or row spaces are contained in $\mathrm{span}(V)$ respectively, and a similar definition holds for $T_{V,W}$ and matrices $V, W$. If $Q$ is a rank $r$ matrix with column space $\mathrm{span}(U)$ and row space $\mathrm{span}(V)$, $T_{U,V}$ is the tangent space at $Q$ with respect to the variety of rank $r$ matrices.

For a tensor $Y$, the *support* of $Y$ refers to the indices corresponding to the non-zero entries of $Y$. Let $\Omega \subseteq [n_1] \times [n_2] \times [n_3]$ denote the support of $Y$. Further, for a slice $Y_i^3$, let $\Omega_i^{(3)} \subseteq [n_1] \times [n_2]$ denote the corresponding sparsity pattern of the slice $Y_i^3$ (more generally $\Omega_i^{(k)}$ can be defined as the sparsity of the matrix resulting from the $i^{th}$ mode $k$ slice). When a tensor contraction of $Y$ is computed along mode $k$, the sparsity of the resulting matrix is the union of the sparsity patterns of each (matrix) slice, i.e. $\Omega^{(k)} = \bigcup_{i=1}^{n_k} \Omega_i^{(k)}$. Let $S\left(\Omega^{(k)}\right)$ denote the set of (sparse) matrices with support $\Omega^{(k)}$. We define the following incoherence parameters:

$$
\zeta(U,V) := \max_{M \in T_{U,V}:\|M\| \le 1} \|M\|_\infty \qquad \zeta(V,W) := \max_{M \in T_{V,W}:\|M\| \le 1} \|M\|_\infty
$$

$$
\mu\left(\Omega^{(k)}\right) := \max_{N \in S\left(\Omega^{(k)}\right):\|N\|_\infty \le 1} \|N\|.
$$

The quantities $\zeta(U,V)$ and $\zeta(V,W)$ being small implies that for contractions of the tensor $Z$, all matrices in the tangent space of those contractions with respect to the variety of rank $r$ matrices are "diffuse", i.e. do not have sparse elements [9]. Similarly, $\mu\left(\Omega^{(k)}\right)$ being small implies that all matrices with the contracted sparsity pattern $\Omega^{(k)}$ are such that their spectrum is "diffuse", i.e. they do not have low rank. We will see specific settings where these forms of incoherence hold for tensors in Section 3.

## 2 Algorithm for Sparse and Low Rank Tensor Decomposition

We now introduce our algorithm to perform sparse and low rank tensor decompositions. We begin with a Lemma:

**Lemma 2.1.** *Let $X \in \mathbb{R}^{n_1 \times n_2 \times n_3}$, with $n_1 \le n_2 \le n_3$ be a tensor of rank $r \le n_1$. Then the rank of $X_a^3$ is at most $r$. Similarly the rank of $X_c^1$ is at most $r$.*

*Proof.* Consider a tensor $X = \sum_{i=1}^r \lambda_i\, u_i \otimes v_i \otimes w_i$. The reader may verify in a straightforward manner that $X_a^3$ enjoys the decomposition:

$$
X_a^3 = \sum_{i=1}^r \lambda_i \langle w_i, a\rangle u_i v_i^T.
\tag{5}
$$

The proof for the rank of $X_c^1$ is analogous. □

Note that while (5) is a matrix decomposition of the contraction, it is not a singular value decomposition (the components need not be orthogonal, for instance). Recovering the factors needs an application of *simultaneous diagonalization*, which we describe next.

**Lemma 2.2.** *[4, 18] Suppose we are given an order 3 tensor $\boldsymbol{X} = \sum_{i=1}^{r} \lambda_i\, u_i \otimes v_i \otimes w_i$ of size $n_1 \times n_2 \times n_3$ satisfying the conditions of Assumption 1.1. Suppose the contractions $X_a^3$ and $X_b^3$ are computed with respect to unit vectors $a, b \in \mathbb{R}^{n_3}$ distributed independently and uniformly on the unit sphere $\mathbb{S}^{n_3-1}$ and consider the matrices $M_1$ and $M_2$ formed as:*

$$M_1 = X_a^3 (X_b^3)^\dagger \qquad M_2 = (X_b^3)^\dagger X_a^3.$$

*Then the eigenvectors of $M_1$ (corresponding to the non-zero eigenvalues) are $\{u_i\}_{i=1,\dots,r}$, and the eigenvectors of $M_2^T$ are $\{v_i\}_{i=1,\dots,r}$.*

**Remark** Note that while the eigenvectors $\{u_i\}$, $\{v_j\}$ are thus determined, a source of ambiguity remains. For a fixed ordering of $\{u_i\}$ one needs to determine the order in which $\{v_j\}$ are to be arranged. This can be (generically) achieved by using the (common) eigenvalues of $M_1$ and $M_2$ for pairing i(f the contractions $X_a^3$, $X_b^3$ are computed with respect to random vectors $a, b$ the eigenvalues are distinct almost surely). Since the eigenvalues of $M_1$, $M_2$ are distinct they can be used to pair the columns of $U$ and $V$.

Lemma 2.2 is essentially a simultaneous diagonalization result [17] that facilitates tensor decomposition [4]. Given a tensor $\boldsymbol{T}$, one can compute two contractions for mode 1 and apply simultaneous diagonalization as described in Lemma 2.2 - this would yield the factors $v_i, w_i$ (up to sign and re-ordering). One can then repeat the same process with mode 3 contractions to obtain $u_i, v_i$. In the final step one can then obtain $\lambda_i$ by solving a system of linear equations. The full algorithm is described in Algorithm 2 in the supplementary material.

For a contraction $Z_v^k$ of a tensor $\boldsymbol{Z}$ with respect to a vector $v$ along mode $k$, consider solving the convex problem:

$$\underset{\mathcal{X},\mathcal{Y}}{\text{minimize}} \qquad \|\mathcal{X}\|_* + \nu_k \|\mathcal{Y}\|_1 \qquad \text{subject to} \qquad Z_v^k = \mathcal{X} + \mathcal{Y}. \qquad (6)$$

Our algorithm, stated in Algorithm 1, proceeds as follows: Given a tensor $\boldsymbol{Z} = \boldsymbol{X} + \boldsymbol{Y}$, we perform two random contractions (w.r.t. vectors $a, b$) of the tensor along mode 3 to obtain matrices $Z_a^{(3)}, Z_b^{(3)}$. Since $\boldsymbol{Z}$ is a sum of sparse and low-rank components, by Lemma 2.1 so are the matrices $Z_a^{(3)}, Z_b^{(3)}$. We thus use (6) to decompose them into constituent sparse and low-rank components, which are the contractions of the matrices $X_a^{(3)}, X_b^{(3)}, Y_a^{(3)}, Y_b^{(3)}$. We then use $X_a^{(3)}, X_b^{(3)}$ and Lemma 2.2 to obtain the factors $U, V$. We perform the same operations along mode 1 to obtain factors $V, W$. In the last step, we solve for the scale factors $\lambda_i$ (a system of linear equations).

Algorithm 2 in the supplementary material, which we adopt for our decomposition problem in Algorithm 1, essentially relies on the idea of simultaneous diagonalization of matrices sharing common row and column spaces [17]. In this paper we do not analyze the situation where random noise is added to all the entries, but only the sparse adversarial noise setting. We note, however, that the key algorithmic insight of using contractions to perform tensor recovery is numerically stable and robust with respect to noise, as has been studied in [4, 11, 17].

Parameters that need to be picked to implement our algorithm are the regularization coefficients $\nu_1, \nu_3$. In the theoretical guarantees we will see that this can be picked in a stable manner, and that a range of values guarantee exact decomposition when the suitable incoherence conditions hold. In practice these coeffients would need to be determined by a cross-validation method. Note also that under suitable random sparsity assumptions [6], the regularization coefficient may be picked to be the inverse of the square-root of the dimension.

## 2.1 Computational Complexity

The computational complexity of our algorithm is dominated by the complexity of perfoming the sparse and low-rank matrix decomposition of the contractions via (6). For simplicity, let us consider

---

**Algorithm 1** Algorithm for sparse and low rank tensor decomposition

---
1: **Input:** Tensor $\mathbf{Z}$, parameters $\nu_1, \nu_3$.
2: Generate contraction vectors $a, b \in \mathbb{R}^{n_3}$ independently and uniformly distributed on unit sphere.
3: Compute mode 3 contractions $Z_a^3$ and $Z_b^3$ respectively.
4: Solve the convex problem (6) with $v = a$, $k = 3$. Call the resulting solution matrices $X_a^3, Y_a^3$, and regularization parameter $\nu_1$.
5: Solve the convex problem (6) with $v = b$, $k = 3$. Call the resulting solution matrices $X_b^3, Y_b^3$ and regularization parameter $\nu_3$.
6: Compute eigen-decomposition of $M_1 := X_a^3(X_b^3)^\dagger$ and $M_2 := (X_b^3)^\dagger X_a^3$. Let $U$ and $V$ denote the matrices whose columns are the eigenvectors of $M_1$ and $M_2^T$ respectively corresponding to the non-zero eigenvalues, in sorted order. (Let $r$ be the (common) rank of $M_1$ and $M_2$.) The eigenvectors, thus arranged are denoted as $\{u_i\}_{i=1,\dots,r}$ and $\{v_i\}_{i=1,\dots,r}$.
7: Generate contraction vectors $c, d \in \mathbb{R}^{n_1}$ independently and uniformly distributed on unit sphere.
8: Solve the convex problem (6) with $v = c$, $k = 1$. Call the resulting solution matrices $X_c^1, Y_c^1$ and regularization parameter $\nu_3$.
9: Solve the convex problem (6) with $v = d$, $k = 1$. Call the resulting solution matrices $X_d^1, Y_d^1$ and regularization parameter $\nu_4$.
10: Compute eigen-decomposition of $M_3 := X_c^1(X_d^1)^\dagger$ and $M_4 := (X_c^1)^\dagger X_d^1$. Let $\tilde{V}$ and $\tilde{W}$ denote the matrices whose columns are the eigenvectors of $M_3$ and $M_4^T$ respectively corresponding to the non-zero eigenvalues, in sorted order. (Let $r$ be the (common) rank of $M_3$ and $M_4$.)
11: Simultaneously reorder the columns of $\tilde{V}, \tilde{W}$, also performing simultaneous sign reversals as necessary so that the columns of $V$ and $\tilde{V}$ are equal, call the resulting matrix $W$ with columns $\{w_i\}_{i=1,\dots,r}$.
12: Solve for $\lambda_i$ in the linear system

$$X_a^3 = \sum_{i=1}^r \lambda_i u_i v_i^T \langle w_i, a \rangle.$$

13: **Output:** Decomposition $\hat{X} := \sum_{i=1}^r \lambda_i \, u_i \otimes v_i \otimes w_i$, $\hat{Y} := \mathbf{Z} - \hat{X}$.

---

the case where the target tensor $\mathbf{Z} \in \mathbb{R}^{n \times n \times n}$ has equal dimensions in different modes. Using a standard first order method, the solution of (6) has a per iteration complexity of $O(n^3)$, and to achieve an accuracy of $\epsilon$, $O\left(\frac{1}{\epsilon}\right)$ iterations are required [22]. Since only four such steps need be performed, the complexity of the method is $O\left(\frac{n^3}{\epsilon}\right)$ where $\epsilon$ is the accuracy to which (6) is solved. Another alternative is to reformulate (6) such that it is amenable to greedy atomic approaches [23], which yields an order of magnitude improvement. We note that in contrast, a tensor unfolding for this problem [14, 21, 26] results in the need to solve much larger convex programs. For instance, for $\mathbf{Z} \in \mathbb{R}^{n \times n \times n}$, the resulting flattened matrix would be of size $n^2 \times n$ and the resulting convex problem would then have a complexity of $O\left(\frac{n^4}{\epsilon}\right)$. For higher order tensors, the gap in computational complexity would increase by further orders of $n$.

## 2.2 Numerical Experiments

We now present numerical results to validate our approach. We perform experiments for tensors of size $50 \times 50 \times 50$ (non-symmetric). A tensor $\mathbf{Z}$ is generated as the sum of a low rank tensor $\mathbf{X}$ and a sparse tensor $\mathbf{Y}$. The low-rank component is generated as follows: Three sets of $r$ unit vecots $u_i, v_i, w_i \in \mathbb{R}^{50}$ are generated randomly, independently and uniformly distributed on the unit sphere. Also a random positive scale factor (uniformly distributed on $[0, 1]$ is chosen and the tensor $\mathbf{X} = \sum_{i=1}^r \lambda_i \, u_i \otimes v_i \times w_i$. The tensor $\mathbf{Y}$ is generated by (Bernoulli) randomly sampling its entries with probability $p$. For each such $p$, we perform 10 trials and apply our algorithm. In all our experiments, the regularization parameter was picked to be $\nu = \frac{1}{\sqrt{n}}$. The optimization problem (6) is solved using CVX in MATLAB. We report success if the MSE is smaller than $10^{-5}$, separately for both the $\mathbf{X}$ and $\mathbf{Y}$ components. We plot the empirical probability of success as a function of $p$ in Fig. 1 (a), (b), for multiple values of the true rank $r$. In Fig. 1 (c), (d) we test the scalability

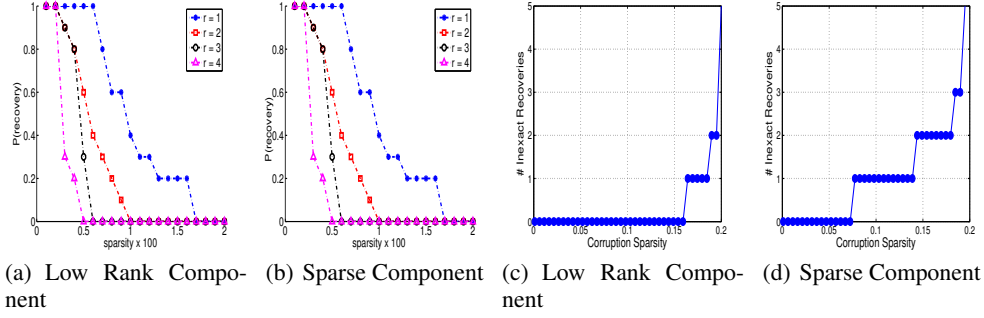

(a) Low Rank Component    (b) Sparse Component    (c) Low Rank Component    (d) Sparse Component

Figure 1: Recovery of the low rank and sparse components from our proposed methods. In figures (a) and (b) we see that the probability of recovery is high when both the rank and sparsity are low. In figures (c) and (d) we study the recovery error for a tensor of dimensions $300 \times 300 \times 300$ and rank 50.

of our method, by generating a random $300 \times 300 \times 300$ tensor of rank 50, and corrupting it with a sparse tensor of varying sparsity level. We run 5 independent trials and see that for low levels of corruption, both the low rank and sparse components are accurately recovered by our method.

## 3 Main Results

We now present the main rigorous guarantees related to the performance of our algorithm. Due to space constraints, the proofs are deferred to the supplementary materials.

**Theorem 3.1.** *Suppose* $Z = X + Y$, *where* $X = \sum_{i=1}^{r} \lambda_i u_i \otimes v_i \otimes w_i$, *has rank* $r \leq n_1$ *and such that the factors satisfy Assumption 1.1. Suppose* $Y$ *has support* $\Omega$ *and the following condition is satisfied:*

$$\mu\left(\Omega^{(3)}\right) \zeta\left(U, V\right) \leq \frac{1}{6} \qquad \mu\left(\Omega^{(1)}\right) \zeta\left(V, W\right) < \frac{1}{6}.$$

*Then Algoritm 1 succeeds in exactly recovering the component tensors, i.e.* $(X, Y) = (\hat{X}, \hat{Y})$ *whenever* $\nu_k$ *are picked so that* $\nu_3 \in \left(\frac{\zeta(U,V)}{1-4\zeta(U,V)\mu(\Omega^{(3)})}, \frac{1-3\zeta(U,V)\mu(\Omega^{(3)})}{\mu(\Omega^{(3)})}\right)$ *and* $\nu_1 \in \left(\frac{\zeta(V,W)}{1-4\zeta(V,W)\mu(\Omega^{(1)})}, \frac{1-3\zeta(V,W)\mu(\Omega^{(1)})}{\mu(\Omega^{(1)})}\right)$. *Specifically, choice of* $\nu_3 = \frac{(3\zeta(U,V))^p}{\left(\mu(\Omega^{(3)})\right)^{1-p}}$ *and* $\nu_1 = \frac{(3\zeta(V,W))^p}{\left(\mu(\Omega^{(1)})\right)^{1-p}}$ *for any* $p \in [0,1]$ *in these respective intervals guarantees exact recovery.*

For a matrix $M$, the degree of $M$, denoted by $\deg(M)$, is the maximum number of non-zeros in any row or column of $M$. For a tensor $Y$, we define the degree along mode $k$, denoted by $\deg_k(Y)$ to be the maximum number of non-zero entries in any row or column of a matrix supported on $\Omega^{(k)}$ (defined in Section 1.2). The degree of $Y$ is denoted by $\deg(Y) := \max_{k \in \{1,2,3\}} \deg_k(Y)$.

**Lemma 3.2.** *We have:*
$$\mu\left(\Omega^{(k)}\right) \leq deg(Y), \text{ for all } k.$$

For a subspace $S \subseteq \mathbb{R}^n$, let us define the incoherence of the subspace as:

$$\beta(S) := \max_i \|P_S e_i\|_2,$$

where $P_S$ denotes the projection operator onto $S$, $e_i$ is a standard unit vector and $\|\cdot\|_2$ is the Euclidean norm of a vector. Let us define:

$$\text{inc}(X) := \max\left\{\beta\left(\text{span}(U)\right), \beta\left(\text{span}(V)\right), \beta\left(\text{span}(W)\right)\right\}$$
$$\text{inc}_3(X) := \max\left\{\beta\left(\text{span}(U)\right), \beta\left(\text{span}(V)\right)\right\}$$
$$\text{inc}_1(X) := \max\left\{\beta\left(\text{span}(V)\right), \beta\left(\text{span}(W)\right)\right\}.$$

Note that $\text{inc}(\boldsymbol{X}) < 1$, always. For many random ensembles of interest, we have that the incoherence scales gracefully with the dimension $n$, i.e.: $\text{inc}(\boldsymbol{X}) \leq K\sqrt{\frac{\max\{r, \log\ n\}}{n}}$.

**Lemma 3.3.** *We have*

$$\zeta\left(U, V\right) \leq 2\ inc(\boldsymbol{X}) \qquad \zeta\left(V, W\right) \leq 2\ inc(\boldsymbol{X}).$$

**Corollary 3.4.** *Let* $\boldsymbol{Z} = \boldsymbol{X} + \boldsymbol{Y}$*, with* $\boldsymbol{X} = \sum_{i=1}^{r} \lambda_i u_i \otimes v_i \otimes w_i$ *and rank* $r \leq n_1$*, the factors satisfy Assumption 1.1 and incoherence* $inc(\boldsymbol{X})$*. Suppose* $\boldsymbol{Y}$ *is sparse and has degree* $deg(\boldsymbol{Y})$*. If the condition*

$$inc(\boldsymbol{X})deg(\boldsymbol{Y}) < \frac{1}{12}$$

*holds then Algorithm 1 successfully recovers the true solution, i.e. .* $(\boldsymbol{X}, \boldsymbol{Y}) = (\hat{\boldsymbol{X}}, \hat{\boldsymbol{Y}})$ *when the parameters*

$$\nu_3 \in \left(\frac{2inc_3(\boldsymbol{X})}{1 - 8deg_3(\boldsymbol{Y})inc_3(\boldsymbol{X})}, \frac{1 - 6deg_3(\boldsymbol{Y})inc_3(\boldsymbol{X})}{deg_3(\boldsymbol{Y})}\right)$$

$$\nu_1 \in \left(\frac{2inc_1(\boldsymbol{X})}{1 - 8deg_1(\boldsymbol{Y})inc_1(\boldsymbol{X})}, \frac{1 - 6deg_1(\boldsymbol{Y})inc_1(\boldsymbol{X})}{deg_1(\boldsymbol{Y})}\right).$$

*Specifically, a choice of* $\nu_3 = \frac{(6inc_3(\boldsymbol{X}))^p}{(2deg_3(\boldsymbol{Y}))^{1-p}}$*,* $\nu_1 = \frac{(6inc_1(\boldsymbol{X}))^p}{(2deg_1(\boldsymbol{Y}))^{1-p}}$ *for any* $p \in [0, 1]$ *is a valid choice that guarantees exact recovery.*

**Remark** Note that Corollary 3.4 presents a *deterministic* guarantee on the recoverability of a sparse corruption of a low rank tensor, and can be viewed as a tensor extension of [9, Corollary 3].

We now consider, for the sake of simplicity, tensors of uniform dimension, i.e. $\boldsymbol{X}, \boldsymbol{Y}, \boldsymbol{Z} \in \mathbb{R}^{n \times n \times n}$. We show that when the low-rank and sparse components are suitably random, the approach outlined in Algorithm 1 achieves exact recovery.

We define the *random sparsity* model to be one where each entry of the tensor $\boldsymbol{Y}$ is non-zero independently and with identical probability $\rho$. We make no assumption about the mangitude of the entries of $\boldsymbol{Y}$, only that its non-zero entries are thus sampled.

**Lemma 3.5.** *Let* $\boldsymbol{X} = \sum_{i=1}^{r} \lambda_i u_i \otimes v_i \otimes w_i$*, where* $u_i, v_i, w_i \in \mathbb{R}^n$ *are uniformly randomly distributed on the unit sphere* $\mathbb{S}^{n-1}$*. Then the incoherence of the tensor* $\boldsymbol{X}$ *satisfies:*

$$inc(\boldsymbol{X}) \leq c_1 \sqrt{\frac{\max\{r, \log\ n\}}{n}}$$

*with probability exceeding* $1 - c_2 n^{-3} \log\ n$ *for some constants* $c_1, c_2$*.*

**Lemma 3.6.** *Suppose the entries of* $\boldsymbol{Y}$ *are sampled according to the random sparsity model, and* $\rho = O\left(\left(n^{\frac{3}{2}} \max(\log n, r)\right)^{-1}\right)$*. Then the tensor* $\boldsymbol{Y}$ *satisfies:* $deg(\boldsymbol{Y}) \leq \frac{\sqrt{n}}{12c_1 \max(\log n, r)}$ *with probability exceeding* $1 - exp\left(-c_3 \frac{\sqrt{n}}{\max(\log n, r)}\right)$ *for some constant* $c_3 > 0$*.*

**Corollary 3.7.** *Let* $\boldsymbol{Z} = \boldsymbol{X} + \boldsymbol{Y}$ *where* $\boldsymbol{X}$ *is low rank with random factors as per the conditions of Lemma 3.5 and* $\boldsymbol{Y}$ *is sparse with random support as per the conditions in Lemma 3.6. Provided* $r \sim o\left(n^{\frac{1}{2}}\right)$*, Algorithm 1 successfully recovers the correct decomposition, i.e.* $(\hat{\boldsymbol{X}}, \hat{\boldsymbol{Y}}) = (\boldsymbol{X}, \boldsymbol{Y})$ *with probability exceeding* $1 - n^{-\alpha}$ *for some* $\alpha > 0$*.*

**Remarks** 1) Under this sampling model, the cardinality of the support of $\boldsymbol{Y}$ is allowed to be as large as $m = O(n^{\frac{3}{2}} \log^{-1} n)$ when the rank $r$ is constant (independent of $n$).

2) We could equivalently have looked at a uniformly random sampling model, i.e. one where a support set of size $m$ is chosen uniformly randomly from the set of all possible support sets of cardinality at most $m$, and our results for exact recovery would have gone through. This follows from the equivalence principle for successful recovery between Bernoulli sampling and uniform sampling, see [6, Appendix 7.1].

3) Note that for the random sparsity ensemble, [6] shows that a choice of $\nu = \frac{1}{\sqrt{n}}$ ensures exact recovery (an additional condition regarding the magnitudes of the factors is needed, however). By extension, the same choice can be shown to work for our setting.

## 4 Extensions

The approach described in Algorithm 1 and the analysis is quite modular and can be adapted to various settings to account for different forms of measurements and robustness models. We do not present an analysis of these situations due to space constraints, but outline how these extensions follow from the current development in a straightforward manner.

1) **Higher Order Tensors:** Algorithm 1 can be extended naturally to the higher order setting. Recall that in the third order case, one needs to recover two contractions along the third mode to discover factors $U, V$ and then two contractions along the first mode to discover factors $V, W$. For an order $K$ tensor of the form $\boldsymbol{Z} \in \mathbb{R}^{n_1 \times \cdots \times n_K}$ which is the sum of a low rank component $\boldsymbol{X} = \sum_{i=1}^{r} \lambda_i \bigotimes_{l=1}^{K} u_i^{(l)}$ and a sparse component $\boldsymbol{Y}$, one needs to compute higher order contractions of $\boldsymbol{Z}$ along $K-1$ different modes. For each of these $K-1$ modes the resulting contraction is the sum of a sparse and low-rank matrix, and thus pairs of matrix problems of the form (6) reveal the sparse and low-rank components of the contractions. The low-rank factors can then be recovered via application of Lemma 2.2 and the full decomposition can thus be recovered. The same guarantees as in Theorem 3.1 and Corollary 3.4 hold verbatim (the notions of incoherence $\mathrm{inc}(\boldsymbol{X})$ and degree $\deg(\boldsymbol{Y})$ of tensors need to be extended to the higher order case in the natural way)

2) **Block sparsity:** Situations where entire slices of the tensor are corrupted may happen in recommender systems with adversarial ratings [10]. A natural approach in this case is to use a convex relaxation of the form

$$\underset{M_1, M_2}{\text{minimize}} \quad \nu_k \|M_1\|_* + \|M_2\|_{1,2}$$
$$\text{subject to} \quad Z_v^k = M_1 + M_2$$

in place of (6) in Algorithm 1. In the above, $\|M\|_{1,2} := \sum_i \|M_i\|_2$, where $M_i$ is the $i^{th}$ column of $M$. Since exact recovery of the block-sparse and low-rank components of the contractions are guaranteed via this relaxation under suitable assumptions [10], the algorithm would inherit associated provable guarantees.

3) **Tensor completion:** In applications such as recommendation systems, it may be desirable to perform tensor *completion* in the presence of sparse corruptions. In [24], an adaptation of Leurgans' algorithm was presented for performing completion from measurements restricted to only four slices of the tensor with near-optimal sample complexity (under suitable genericity assumptions about the tensor). We note that it is straightforward to blend Algorithm 1 with this method to achieve completion with sparse corruptions. Recalling that $\boldsymbol{Z} = \boldsymbol{X} + \boldsymbol{Y}$ and therefore $Z_k^3 = X_k^3 + Y_k^3$ (i.e. the $k^{th}$ mode 3 slice of $\boldsymbol{Z}$ is a sum of sparse and low rank slices of $\boldsymbol{X}$ and $\boldsymbol{Y}$), if only a subset of elements of $Z_k^3$ (say $P_\Lambda\left(Z_k^3\right)$) is observed for some index set $\Lambda$, we can replace (6) in Algorithm 1 with

$$\underset{M_1, M_2}{\text{minimize}} \quad \nu_k \|M_1\|_* + \|M_2\|_1 \qquad \text{subject to} \qquad P_\Lambda\left(Z_v^k\right) = P_\Lambda\left(M_1 + M_2\right).$$

Under suitable incoherence assumptions [6, Theorem 1.2], the above will achieve exact recovery of the slices. Once four slices are accurately recovered, one can then use Leurgans' algorithm to recover the full tensor [24, Theorem 3.6]. Indeed the above idea can be extended more generally to the concept of deconvolving a sum of sparse and low-rank tensors from separable measurements [24].

4) **Non-convex approaches:** A basic primitive for sparse and low-rank tensor decomposition used in this paper is that of using (6) for matrix decomposition. More efficient *non-convex* approaches such as the ones described in [22] may be used instead to speed up Algorithm 1. These alternative nonconvex methods [22] requre $O(rn^2)$ steps per iterations, and $O\left(\log \frac{1}{\epsilon}\right)$ iterations resulting in a total complexity of $O\left(rn^2 \log \frac{1}{\epsilon}\right)$ for solving the decomposition of the contractions to an accuracy of $\epsilon$.

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
