[Supplementary Material]

# Sparse and Low-Rank Tensor Decomposition: Supplementary Material

## 1 Leurgans' Algorithm

For the sake of completeness, we present Leurgans' algorithm for tensor decomposition. The algorithm essentially uses simultaneous diagonalization (Lemma 2.2) at its core.

---
**Algorithm 2** Leurgans' algorithm for tensor decomposition

---
1: **Input:** Tensor $\boldsymbol{X}$
2: Generate contraction vectors $a, b \in \mathbb{R}^{n_3}$, $c, d \in \mathbb{R}^{n_1}$ uniformly randomly distributed on the unit sphere.
3: Compute mode 3 contractions $X_a^3$ and $X_b^3$ respectively.
4: Compute eigen-decomposition of $M_1 := X_a^3 (X_b^3)^\dagger$ and $M_2 := (X_b^3)^\dagger X_a$. Let $U$ and $V$ denote the matrices whose columns are the eigenvectors of $M_1$ and $M_2^T$ respectively corresponding to the non-zero eigenvalues, in sorted order. (Let $r$ be the (common) rank of $M_1$ and $M_2$.) The eigenvectors, thus arranged are denoted as $\{u_i\}_{i=1,\ldots,r}$ and $\{v_i\}_{i=1,\ldots,r}$.
5: Compute mode 1 contractions $X_c^1$ and $X_d^1$ respectively.
6: Compute eigen-decomposition of $M_3 := X_c^1 (X_d^1)^\dagger$ and $M_4 := (X_c^1)^\dagger X_d^1$. Let $\tilde{V}$ and $\tilde{W}$ denote the matrices whose columns are the eigenvectors of $M_3$ and $M_4^T$ respectively corresponding to the non-zero eigenvalues, in sorted order. (Let $r$ be the (common) rank of $M_3$ and $M_4$.)
7: Simultaneously reorder the columns of $\tilde{V}, \tilde{W}$, also performing simultaneous sign reversals as necessary so that the columns of $V$ and $\tilde{V}$ are equal, call the resulting matrix $W$ with columns $\{w_i\}_{i=1,\ldots,r}$.
8: Solve for $\lambda_i$ in the linear system

$$\boldsymbol{X} = \sum_{i=1}^{r} \lambda_i u_i \otimes v_i \otimes w_i.$$

9: **Output:** Decomposition $\boldsymbol{X} = \sum_{i=1}^{r} \lambda_i \, u_i \otimes v_i \otimes w_i$.

---

## 2 Proofs

Since our algorithmic approach reduces the tensor decomposition problem to that of sparse and low-rank matrix decomposition, some of the proofs of the lemmas below reuse existing results. Rather than reproving the intermediate results here, we simply refer the reader to the appropriate references.

Our first lemma establishes that given information about two contractions of a tensor, one may recover the the tensor via linear algebraic operations.

**Lemma 2.2.** *[4, 19] Suppose we are given an order 3 tensor $\boldsymbol{X} = \sum_{i=1}^{r} \lambda_i \, u_i \otimes v_i \otimes w_i$ of size $n_1 \times n_2 \times n_3$ satisfying the conditions of Assumption 1.1. Suppose the contractions $X_a^3$ and $X_b^3$ are computed with respect to unit*

vectors $a, b \in \mathbb{R}^{n_3}$ distributed independently and uniformly on the unit sphere $\mathbb{S}^{n_3-1}$ and consider the matrices $M_1$ and $M_2$ formed as:

$$M_1 = X_a^3 (X_b^3)^\dagger \qquad M_2 = (X_b^3)^\dagger X_a^3.$$

Then the eigenvectors of $M_1$ (corresponding to the non-zero eigenvalues) are $\{u_i\}_{i=1,\dots,r}$, and the eigenvectors of $M_2^T$ are $\{v_i\}_{i=1,\dots,r}$.

*Proof.* Suppose we are given an order 3 tensor $\boldsymbol{X} = \sum_{i=1}^r \lambda_i \, u_i \otimes v_i \otimes w_i \in \mathbb{R}^{n_1 \times n_2 \times n_3}$. From the definition of contraction (2), it is straightforward to see that

$$X_a^3 = U D_a V^T \qquad D_a = \mathrm{diag}(\lambda_1 a^T w_1, \dots, \lambda_r a^T w_r)$$

$$X_b^3 = U D_b V^T \qquad D_b = \mathrm{diag}(\lambda_1 b^T w_1, \dots, \lambda_r b^T w_r).$$

In the above decompositions, $U \in \mathbb{R}^{n_1 \times r}$, $V \in \mathbb{R}^{n_2 \times r}$, and the matrices $D_a, D_b \in \mathbb{R}^{r \times r}$ are diagonal and non-singular almost surely (since $a$, $b$ are random). Now,

$$\begin{aligned} M_1 &:= X_a^3 (X_b^3)^\dagger \\ &= U D_a V^T (V^\dagger)^T D_b^{-1} U^\dagger \\ &= U D_a D_b^{-1} U^\dagger \end{aligned} \tag{7}$$

and similarly we obtain

$$M_2^T = V D_b^{-1} D_a V^\dagger. \tag{8}$$

Since we have $M_1 U = U D_a D_b^{-1}$ and $M_2^T V = V D_b^{-1} D_a$, it follows that the columns of $U$ and $V$ are eigenvectors of $M_1$ and $M_2^T$ respectively (with corresponding eigenvalues given by the diagonal matrices $D_a D_b^{-1}$ and $D_b^{-1} D_a$). $\square$

**Theorem 3.1.** *Suppose $\boldsymbol{Z} = \boldsymbol{X} + \boldsymbol{Y}$, where $\boldsymbol{X} = \sum_{i=1}^r \lambda_i u_i \otimes v_i \otimes w_i$, has rank $r \leq n_1$ and such that the factors satisfy Assumption 1.1. Suppose $\boldsymbol{Y}$ has support $\Omega$ and the following condition is satisfied.*

$$\mu\left(\Omega^{(3)}\right) \zeta(U, V) \leq \frac{1}{6} \qquad \mu\left(\Omega^{(1)}\right) \zeta(V, W) < \frac{1}{6}.$$

*Then Algorithm 1 succeeds in exactly recovering the component tensors, i.e. $(\boldsymbol{X}, \boldsymbol{Y}) = (\hat{\boldsymbol{X}}, \hat{\boldsymbol{Y}})$ whenever $\nu_k$ are picked so that $\nu_3 \in \left( \frac{\zeta(U,V)}{1 - 4\zeta(U,V)\mu(\Omega^{(3)})}, \frac{1 - 3\zeta(U,V)\mu(\Omega^{(3)})}{\mu(\Omega^{(3)})} \right)$ and $\nu_1 \in \left( \frac{\zeta(V,W)}{1 - 4\zeta(V,W)\mu(\Omega^{(1)})}, \frac{1 - 3\zeta(V,W)\mu(\Omega^{(1)})}{\mu(\Omega^{(1)})} \right)$. Specifically, choice of $\nu_3 = \frac{(3\zeta(U,V))^p}{(\mu(\Omega^{(3)}))^{1-p}}$ and $\nu_1 = \frac{(3\zeta(V,W))^p}{(\mu(\Omega^{(1)}))^{1-p}}$ for any $p \in [0,1]$ in these respective intervals guarantees exact recovery.*

*Proof.* Since $\boldsymbol{Z} = \boldsymbol{X} + \boldsymbol{Y}$, we have $Z_a^3 = X_a^3 + Y_a^3$. By Lemma 2.1 $X_a^3$ is a low-rank matrix with row space $\mathrm{span}(U)$ and column space $\mathrm{span}(V)$. Hence the incoherence parameter for $X_a^3$ is precisely $\zeta(U, V)$. Since $\boldsymbol{Y}$ is sparse with support $\Omega$, $Y_a^3$ is sparse with support $\Omega^{(3)}$. By assumption, $\mu(\Omega^{(3)}) \zeta(U, V) \leq \frac{1}{6}$. By [10, Theorem 2], the convex relaxation (6) with the prescribed regularization parameter exactly recovers the unique low-rank and sparse components, i.e. $X_a^3, Y_a^3$. Similarly, the procedure repeated with respect to the contraction vector $b$ recovers $X_b^3$. By Lemma 2.2, step 6 of Algorithm 1 exactly recovers the $U, V$. The same procedure repeated with contractions along the first mode with respect to $c, d$ ensures recovery of $V, W$. Note that in Step 12 of Algorithm 1, the linear system is full rank (since the factors are linearly independent by Assumption 1.1), overdetermined and thus has a unique and correct solution. $\square$

**Lemma 3.2.** *We have:*

$$\mu\left(\Omega^{(k)}\right) \leq deg(\boldsymbol{Y}), \text{ for all } k.$$

*Proof.* Given a tensor $\boldsymbol{Y}$ with support $\Omega$, the sparsity pattern of $Y_a^3$ is contained within $\Omega^{(3)}$. By the definition of the degree of $\boldsymbol{Y}$, we have $\deg(Y_a^3) \leq \deg(\boldsymbol{Y})$. By [10, Proposition 3] the result follows. $\square$

**Lemma 3.3.** *We have*

$$\zeta\left(U, V\right) \leq 2inc(\boldsymbol{X}) \qquad \zeta\left(V, W\right) \leq 2inc(\boldsymbol{X}).$$

*Proof.* From [10, Proposition 4], we have $\zeta\left(U, V\right) \leq 2 \max\left\{\beta\left(\mathrm{span}(U)\right), \beta\left(\mathrm{span}(V)\right)\right\}$. Similarly, we have $\zeta\left(V, W\right) \leq 2 \max\left\{\beta\left(\mathrm{span}(V)\right), \beta\left(\mathrm{span}(W)\right)\right\}$. The result follows by applying the definition of $inc(\boldsymbol{X})$. $\qquad\square$

**Corollary 3.4.** *Let $\boldsymbol{Z} = \boldsymbol{X} + \boldsymbol{Y}$, with $\boldsymbol{X} = \sum_{i=1}^{r} \lambda_i u_i \otimes v_i \otimes w_i$ and rank $r \leq n_1$, the factors satisfy Assumption 1.1 and incoherence $inc(\boldsymbol{X})$. Suppose $\boldsymbol{Y}$ is sparse and has degree $deg(\boldsymbol{Y})$. If the condition*

$$inc(\boldsymbol{X})deg(\boldsymbol{Y}) < \frac{1}{12}$$

*holds then Algorithm 1 successfully recovers the true solution, i.e. . $(\boldsymbol{X}, \boldsymbol{Y}) = (\hat{\boldsymbol{X}}, \hat{\boldsymbol{Y}})$ when the parameters*

$$\nu_3 \in \left(\frac{2inc_3(\boldsymbol{X})}{1 - 8deg_3(\boldsymbol{Y})inc_3(\boldsymbol{X})}, \frac{1 - 6deg_3(\boldsymbol{Y})inc_3(\boldsymbol{X})}{deg_3(\boldsymbol{Y})}\right)$$

$$\nu_1 \in \left(\frac{2inc_1(\boldsymbol{X})}{1 - 8deg_1(\boldsymbol{Y})inc_1(\boldsymbol{X})}, \frac{1 - 6deg_1(\boldsymbol{Y})inc_1(\boldsymbol{X})}{deg_1(\boldsymbol{Y})}\right).$$

*Specifically, a choice of $\nu_3 = \frac{(6inc_3(\boldsymbol{X}))^p}{(2deg_3(\boldsymbol{Y}))^{1-p}}$, $\nu_1 = \frac{(6inc_1(\boldsymbol{X}))^p}{(2deg_1(\boldsymbol{Y}))^{1-p}}$ for any $p \in [0, 1]$ is a valid choice that guarantees exact recovery.*

*Proof.* Follows immediately from Lemma 3.2, Lemma 3.3, and the conditions of Theorem 3.1 being satisfied. $\qquad\square$

We consider, for the sake of simplicity, tensors of uniform dimension, i.e. $\boldsymbol{X}, \boldsymbol{Y}, \boldsymbol{Z} \in \mathbb{R}^{n \times n \times n}$. We define the *random sparsity* model to be one where each entry of the tensor $\boldsymbol{Y}$ is non-zero independently and with identical probability $\rho$. We make no assumption about the mangitude of the entries of $\boldsymbol{Y}$, only that its non-zero entries are thus sampled.

**Lemma 3.5.** *Let $\boldsymbol{X} = \sum_{i=1}^{r} \lambda_i u_i \otimes v_i \otimes w_i$, where $u_i, v_i, w_i \in \mathbb{R}^n$ are uniformly randomly distributed on the unit sphere $\mathbb{S}^{n-1}$. Then the incoherence of the tensor $\boldsymbol{X}$ satisifies:*

$$inc(\boldsymbol{X}) \leq c_1 \sqrt{\frac{\max\left\{r, \log n\right\}}{n}}$$

*for some constants $c_1, c_2$, with probability exceeding $1 - c_2 n^{-3} \log n$.*

*Proof.* Since $u_i$ are picked uniformly randomly on the unit sphere, the subspace $\mathrm{span}(U)$ is a uniformly random subspace. Equivalently, $\mathrm{span}(U) = \mathrm{span}(\tilde{U})$ for some random matrix $\tilde{U}$ which is uniformly distributed with respect to the set of partial isometries in $\mathbb{R}^{n \times k}$. By [7, Lemma 2.2] we have that

$$\beta\left(\mathrm{span}(U)\right) \leq c_1 \sqrt{\frac{\max\left\{r, \log n\right\}}{n}}$$

with probability exceeding $1 - k_0 n^{-3} \log n$ for some constant $k_0$. The same results hold for the incoherences of $\mathrm{span}(V), \mathrm{span}(W)$. By the definition of $inc(\boldsymbol{X})$, we have the required result. required result. $\qquad\square$

**Lemma 3.6.** *Suppose the entries of $\boldsymbol{Y}$ are sampled according to the random sparsity model, and*

$$\rho = O\left(\left(n^{\frac{3}{2}} \max(\log n, r)\right)^{-1}\right).$$

*Then the tensor $\boldsymbol{Y}$ satisfies:*

$$deg(\boldsymbol{Y}) \leq \frac{\sqrt{n}}{12c_1 \max(\log n, r)}$$

*with probability exceeding $1 - exp\left(-c_3 \frac{\sqrt{n}}{\max(\log n, r)}\right)$ for some constant $c_3 > 0$.*

*Proof.* To bound $\deg(\boldsymbol{Y})$ we must bound the degree of any matrix supported on $\Omega^{(k)}$ for $k = 1, 2, 3$. To this end we introduce the following sets of random variables:

- Let $B_{ijk} \sim \text{Bernoulli}(\rho)$ be the random variable such that $B_{ijk} = 1$ when $(i, j, k) \in \Omega$ and 0 otherwise.

- Let $C$ be a matrix such that $C_{ij} = 1$ if $(i, j) \in \Omega^{(3)}$ and 0 otherwise.

We have that

$$C_{ij} \leq \sum_{k=1}^{n} B_{ijk}.$$

Hence, for any column of $C$ (say $j^{th}$ column), we have that the number of non-zeroes in the column (let us denote this by $\deg(C_j)$) is given by:

$$\deg(C_j) = \sum_{i=1}^{n} C_{ij} \leq \sum_{i=1}^{n} \sum_{k=1}^{n} B_{ijk}. \tag{9}$$

Since (9) is a sum of $i.i.d.$ Bernoulli random variables, we have by the (multiplicative form of) Chernoff-Hoeffding inequality:

$$\mathbb{P}\left(\deg(C_j) > 2n^2\rho\right) \leq \exp\left(-c_0 \frac{\sqrt{n}}{\max(\log n, r)}\right)$$

for some constant $c_0$. In other words,

$$\mathbb{P}\left(\deg(C_j) > \frac{\sqrt{n}}{12c_1 \max(\log n, r)}\right) \leq \exp\left(-c_0 \frac{\sqrt{n}}{\max(\log n, r)}\right).$$

The same argument applies for all the rows and columns of $C$, and thus the same bound applies. By taking a union bounds over these rows and columns we have that:

$$\mathbb{P}\left(\deg(C) > \frac{\sqrt{n}}{12c_1 \max(\log n, r)}\right) \leq 2n \exp\left(-c_2 \frac{\sqrt{n}}{\max(\log n, r)}\right)$$

for some constant $c_2$. Note that $\deg(C)$ is an upper bound on $\deg_3(\boldsymbol{X})$. In an identical manner, we can bound the degrees along the first and second mode, and taking union bounds over the three modes we get the result. $\square$

**Corollary 3.7.** *Let $\boldsymbol{Z} = \boldsymbol{X} + \boldsymbol{Y}$ where $\boldsymbol{X}$ is low rank with random factors as per the conditions of Lemma 3.5 and $\boldsymbol{Y}$ is sparse with random support as per the conditions in Lemma 3.6. Provided $r \sim o\left(n^{\frac{1}{2}}\right)$, Algorithm 1 successfully recovers the correct decomposition, i.e. $(\hat{\boldsymbol{X}}, \hat{\boldsymbol{Y}}) = (\boldsymbol{X}, \boldsymbol{Y})$ with probability exceeding $1 - n^{-\alpha}$ for some $\alpha > 0$.*

*Proof.* The result follows immediately from Lemma 3.5, Lemma 3.6, and Corollary 3.4. $\square$