[Reviews · NeurIPS 2015]

Submitted by Assigned_Reviewer_1

In this paper, a modified Leurgans' algorithm is proposed to do decompositions for sparse and low rank tensors. Tensor decomposition is very hard and usually computationally expensive. So it is interesting to study efficient algorithms to do tensor decompsitions for sparse and low rank tensors. In the paper, the authors use order 3 tensors as examples, and clearly present the algorithm and theoretically discuss the efficiency. In section 4, the authors discuss the algorithm can be extended to higher order tensor. The paper is well organized. Here are some questions:

(1) In the algorithm, it needs to do several eigenvalue decompositions for matrices, which are computationally expensive. And, it also needs to solve two convex matrix optimization problems (6). Can this method be applied to solve large scale tensor decompositions? How to understand the sparsity in the paper? What is the largest tensor that can be decomposed by this algorithm? In section 2,2, Numerical examples only present results for 50 dimensional order-3 tensor, which is not enough to show the efficiency of the algorithm.

(2) Page 2, line 071, the definition of low-rank is r <= n1. In Numerical implementation part, (Section 2.2) for 50 *50 *50 Tensor, computational results are list for r = 1-4, which is very small. What is the computational result for r =40-49? Is it also very good, or the algorithm only works for very small r?

(3) Section 2.2, how to solve problem (6) in your implementation?

Other suggestions:

- Typo, line 093, the resulting n*n matrix, should be n1*n2 matrix;

- Typo, line 96-98, \sum_k should be \sum_i;

- Typo, page 5, line 216, X_a should be X_a^3;

- Typo, page 4, line 198, "coefficient" should be coefficient.
Summary: In this paper, a modified Leurgans' algorithm is proposed to do decompositions for sparse and low rank tensors. Tensor decomposition is very hard and usually computationally expensive. So it is interesting to study efficient algorithms to do tensor decompositions for sparse and low rank tensors. The paper is organized well. However, the numerical implementation part is not good enough. It only shows the result for a very special order-3 tensor with very small r. It might be better to give more examples.

Submitted by Assigned_Reviewer_2

This paper proposes a tensor decomposition algorithm with a sparse noise setting. The proposed method is interesting and useful for general settings of tensor decompositions. And the algorithm could be tractable by reducing the original problem into matrix decompositions of matrices in pairs of modes. However experimental evaluation is limited.

Quality: This result is of decent quality. The proposed approach is general and useful, which can easily extend to high order tensors without increasing much computational cost. However numerical evaluation is limited and the used evaluation measure is unusual. The reason why authors used the accurate probability instead of MSE should be provided. Additional numerical evaluations with several noise settings are necessary.

Clarity: This paper totally reads very well.

However no optimization algorithm for (6) is provided. I guess authors used [22] or [23]. At least, the summary of used algorithms should be described. In section 2.2, the model description X=\sum_{i=1}^r \lambda ... includes a mistake.

Originality: The proposed decomposition algorithms and its theoretical analysis are quite novel.

Significance: The proposed approach seems to be a promising decomposition. This approach can be easily implemented and extended to more general settings of tensor decomposition. To stress these advantages, discussions about related works is necessary.

Summary: The proposed tensor decomposition algorithm is interesting and useful.

Submitted by Assigned_Reviewer_3

Summary: It is firstly assumed that an observable tensor consists of two additive parts: a low-rank tensor that is not sparse; and a sparse corruption tensor that is not low-rank. Then the paper proposes an extension of the Leurgans' Algorithm in terms of solving the convex problem defined by the model assumption. This extension has the advantage of simplicity and less computational complexity. With theoretical proofs provided afterwards it can be verified that the both assumed tensor components can be recovered correctly under given conditions.

Quality: The contents are very well structured and the main idea is clearly introduced and motivated. The provided proofs are also convincing and helpful for understanding. But there are quite a few number of trivial typos. Clarity: Altogether the main concept and motivation are quite comprehensible. The authors have made especially remarkable effort to derive and demonstrate the constraints of the proposed approach.

Originality: Although the proposed algorithm strongly depends on that of Leurgans, its combination with convex optimization could be a novel solution to the defined assumption.

Significance: It has been shown that the model is capable of modeling the two additive tensors being low-rank and sparse, respectively. But the proposed algorithm has certain constraints that might compromise the applicability and significance. Especially the final conclusion (Corollary 3.7) is based on uniform dimension without giving further arguments.
Summary: The paper proposes an interesting extension of the Leurgans' Algorithm. The concept and main results are very clearly formulated, although it is yet unclear whether the applicability of the model might suffer from the mentioned constraints, since experiments on real-life data are missing in the paper.

Submitted by Assigned_Reviewer_4

In this submission, the authors propose to solve a tensor factorisation problem of the form Z=X+Y, where X is a low-rank representation of Z and Y represents the sparse corruption. The authors define two subspaces of matrices T_{U,V} and T_{V,W} for a third-order non-symmetric tensor that describe the span of tensor after contraction along either the 1st or 3d mode, respectively. The sparsity after contraction along mode m is characterised by the union of support sets of each matrix (tensor slice) extracted along this mode. Therefore, the authors define incoherence parameters based on L_{inf} norms to control the sparsity, e.g. make spectra of matrices 'diffuse' rather than sparse. The authors employ an extension of the simultaneous diagonalisation [18] by contracting a tensor along modes 1 and 3 by vectors a and b randomly chosen for uniformly distributed unit sphere. This allows obtaining a set of eigenvalues u and v, and v' and w. After pairing eigenvalues u, v, and w, a set of equation arises in which the authors can solve for eigenvectors. For simultaneously imposing low-rank X and sparse Y, the authors apply the simultaneous diagonalisation on contractions of X and Y, respectively, and impose the nuclear norm on all X and \ell_1 norm on all Y, and solve equation 6. This particular subproblem is well explained in [10], e.g. equation 1.3.

For evaluations, the authors generate 50^3 dimensional non-symmetric tensors Z composed from low rank X and sparse Y according to distributions described in the paper. The authors present the likelihood of recovery of tensors X and Y from Z w.r.t sparsity and true tensor rank r.

Pros: - an interesting non-symmetric tensor factorisation/decomposition problem - interesting expansion of existing symultaneous positive-definite matrix diagonalisation problem [18]. - interesting way of imposing the low-rank and sparsity related constraints on contractions extending problem [10]

Cons: - this submission would benefit from more evaluations, e.g. beyond figure 1 - the last steps of the algorithm in section 2 could be described better - the proofs feel a little bit hard to follow.

Major comments: 1. This is a very interesting paper that extends several ideas verging from simultaneous diagonalisation for a set of matrices to low-rank and sparse matrix recovery by applying/extending these methods to third order non-symmetric tensors. The maths behind these methods seems convincing.

2. The symultaneous diagonalisation in [18] identifies the problem with using merely one projection vector (equvalent of one contraction). The authors of [18] note in section 3 that the quality of estimated eigenvalues depends on the value of the smallest eigengap. Hence, the authors of [18] propose random projections in order to minimise the situations where a single projection could lead to zeroing some eigenvalues. Do authors of this submission perform multiple contractions? It feels so however it is still very unclear form section 2 or algorithm 1. How does this affect the stability of diagonalisaton given non-symmetric tensors require solving for three sets of eigenvectors?

3. The evaluations of the proposed algorithm feel somewhat limited. It'd be very convincing to see an experiment on non-synthetic data. Also, some of the guarantees

given in section 3 could be also illustrated in the experimental setting. For instance, what happens in practical simulations when v_3 and v_1 are outside of the guaranteed regime from corollary 3.4?

This paper should be interesting to the community researching higher-order tensors. It is sufficiently clear and original despite it relies heavily on prior developments.
Summary: In this submission, the authors propose to solve a tensor factorisation problem of the form Z=X+Y, where X is a low-rank representation of Z and Y represents the sparse corruption.

Author Feedback
Author rebuttal: We thank the reviewers for their feedback.

After a careful reading of the feedback, all reviewers seem to agree that while the problem formulation, the algorithmic approach and the theory are interesting, more numerical evidence will improve the paper.

The initial objective of the authors was to lay down a theoretical foundation for decomposition in the presence of corruptions, generalizing [10,6]. We presented just basic numerics to support the theory. We see now that more numerics will greatly strengthen the paper. Doing so is quite straightforward and we plan to include the following concrete experiments:

1. Decomposition of 500x500x500 tensor of rank 50 + sparse tensor using specialized methods such as [ALM], [22] with varying noise-levels and MSE plots.

2. Experiment involving foreground/background separation using real-world video data(generalizing image separation described in [6]). Sparse components can be thought of as foreground objects and low-rank component as a slowly moving background.

Additionally, we would like to respond to other specific points:

Reviewer 1
"..used the accurate probability instead of MSE should be provided."
Plots such as these (phase transition plots) are standard in the literature on model selection (e.g. compressed sensing, matrix completion, and Robust PCA) when exact recovery is possible. To study the regime where exact recovery is possible, phase transition plots of probabilities are convenient. However, MSE plots will be added.

"..the summary of used algorithms should be described"
We used CVX to solve the (convex) sparse/low rank matrix problem. We will describe this in the paper.

Reviewer 2
"..last steps of the algorithm in section 2 could be described better"
We will rephrase things more clearly.

"..proofs feel a little bit hard to follow."
We will expand/simplify the proofs in the supplementary section so that they are easier to follow.

"..perform multiple contractions? How does this affect the stability of diagonalisaton ..?"
Yes we perform two contractions along each mode. Contractions are needed for two of the three modes for a third order tensor. Each contraction yields two of the three sets of eigenvectors. In the presence of sparse arbitrary corruptions, the method is provably stable. While this paper does not deal with dense (suitably random) corruptions, the robustness can be established by extending techniques in [4,18] to our setting.
We will clarify this point in the paper.

"..what happens when v_3 and v_1 are outside of the guaranteed regime ...?"
We will conduct additional experiments and report.

Reviewer 4
" ..(Corollary 3.7) is based on uniform dimension without giving further arguments."
Corollary 3.7 follows by a routine calculation. We will expand in the supplementary material.

Reviewer 5
" ..The presentation needs substantial improvement ..."
We hope that the following changes will be satisfactory.
a. Additional experiments as outlined above (b) Fixing typos (c) Expansion of proofs where terse (d) Improving parts of description of the algorithm

Reviewer 6
"How to understand the sparsity in the paper?"
In Robust Principal Component Analysis, Low rank component X is the signal of interest. Sparse component Y represents a few but arbitrary corruptions to the data, in the spirit of sparse PCA, extended to tensors. See lines 37-42 of paper. In video applications, the sparse component may correspond to the activity of a small foreground object, and the low rank component as a slowly-changing background scene.

"..largest tensor that can be decomposed by this algorithm?"
This depends on the computational resources available, and is best characterized by the computational complexity discussion in Sec. 2.1. On a laptop computer one can potentially solve 3000 dimensional instances of rank 50, where the bottleneck would be storage of the tensor.

"...What is the computational result for r =40-49? ...algorithm only works for very small r?"
Fundamentally the decomposition into components (either in the tensor setting discussed here or the matrix setting discussed in [6,12]) is possible only when one of the components is sparse and the other low-rank. In the absence of this assumption, decomposition is not possible by any algorithm.
Accordingly, both our theory (Lemma 3.6), and experiments confirm that as the rank increases, fewer and fewer sparse corruptions are permissible in order to separate the sparse and low rank components.
That said, we plan to have experiments for higher values of r (r~sqrt(n)) and moderate levels of sparse corruption.

"...how to solve problem (6) in your implementation?"
This is a standard convex optimization problem, for which we used CVX in MATLAB. For larger experiments, we will use special purpose solvers [ALM], [22].

=
[ALM] "The Augmented Lagrange Multiplier Method for Exact Recovery of Corrupted Low-Rank Matrices", Z. Lin, M. Chen and Y. Ma